# The shifting landscape of vaccine discourse: Insights from a decade of pre- to post-COVID-19 vaccine posts on social media

**Nikesh Gyawali**[1], **Doina Caragea**[1]*, **Cornelia Caragea**[2], **Saif M. Mohammad**[3]

**1** Kansas State University, Manhattan, Kansas, United States of America, **2** University of Illinois, Chicago, Illinois, United States of America, **3** National Research Council Canada, Ottawa, Ontario, Canada

* dcaragea@ksu.edu

## Abstract

In this work, we study English-language vaccine discourse in social media posts, specifically posts on X (formerly Twitter), in seven years before the COVID-19 outbreak (2013 to 2019) and three years after the outbreak was first reported (2020 to 2022). Drawing on theories from social cognition and the stereotype content model in Social Psychology, we analyze how English speakers talk about vaccines on social media to understand the evolving narrative around vaccines in social media posts. To do that, we first introduce a novel dataset comprising 18.7 million curated posts on vaccine discourse from 2013 to 2022. This extensive collection—filtered down from an initial 129 million posts through rigorous preprocessing—captures both pre-COVID and COVID-19 periods, offering valuable insights into the evolution of English-speaking X users' perceptions related to vaccines. Our analysis shows that the COVID-19 pandemic led to complex shifts in X users' sentiment and discourse around vaccines. We observe that negative emotion word usage decreased during the pandemic, with notable rises in usage of surprise, and trust related emotion words. Furthermore, vaccine-related language tended to use more warmth-focused words associated with trustworthiness, along with positive, competence-focused words during the early days of the pandemic, with a marked rise in negative word usage towards the end of the pandemic, possibly reflecting a growing vaccine hesitancy and skepticism.

## Introduction

Vaccines have been one of the most significant inventions in the history of mankind, and during the twentieth century, they helped eliminate most of the childhood diseases that were causing millions of deaths every year [1]. In the twenty-first century, vaccines still play a major part in safeguarding people's health from emerging infectious diseases, especially for vulnerable populations in low-income countries [1]. On

**Data availability statement:** The dataset and code for this study are publicly available from the GitHub repository (https://github.com/gnikesh/vaccine-dataset).

**Funding:** This study was partially financially supported by the Cognitive and Neurobiological Approaches to Plasticity (CNAP) Center of Biomedical Research Excellence (COBRE) of the National Institutes of Health in the form of a grant (P20GM113109) received by DC. No additional external funding was received for this study. The content is solely the responsibility of the authors and does not necessarily represent the official views of the National Institutes of Health.

**Competing interests:** The authors have declared that no competing interests exist.

March 11, 2020, the World Health Organization (WHO) declared the novel coronavirus (COVID-19) outbreak a global pandemic, bringing the attention of the world to vaccines like never before. But with that, we have also seen a manifold increase in vaccine concerns and polarization of opinions. Thus, a deeper understanding of public perceptions of vaccines is crucial for devising effective communication strategies by health organizations.

X (formerly Twitter) has been one of the most engaging and influential microblogging platforms over the decades, allowing users to post and read short 280-character messages called posts (formerly tweets). As such, X has been regarded as a hub for social, political, and even health-related discussions. In this work, we study temporal patterns in English-language vaccine discourse on X, aiming to understand how such patterns have changed over the years, especially across the pre-COVID-19 (before 2020) and COVID-19 years (2020 to 2022). While extensive research exists on vaccine discourse, a critical gap remains in the availability of large-scale, high-quality datasets that track the evolution of public perceptions over time. Towards this goal, our study introduces a dataset consisting of 18.7 million meticulously curated English-language posts from a major social media platform, spanning a decade of discourse. In addition to introducing this large volume of unlabeled data, we also utilize theoretically grounded social cognition theory to examine the X users' perceptions regarding vaccines. This resource not only enables a comprehensive analysis of shifts in vaccine perceptions both before and during the COVID-19 pandemic but also serves as a valuable platform for advancing machine-learning methodologies. Specifically, the large volume of unlabeled data presents a promising opportunity to explore modern semi-supervised learning techniques, uncover deeper insights, and enhance automated annotation processes. Despite limitations on access to X posts since 2023, the vaccine-related X posts remain a significant and unique source of information to examine public perceptions. Other sources of information (such as surveys or even Reddit posts about vaccines) tend to be smaller, and while they are interesting in their own right, they do not make studying X posts redundant.

Our study draws on two complementary theoretical perspectives from social psychology and cognition theory to interpret large-scale English-language vaccine discourse on X.

**Emotion dynamics:** We apply emotion dynamics theory, which views emotions as dynamic, socially regulated processes that fluctuate across time and context. In psychology, emotion dynamics refers to the study of patterns of change and regularity in emotion [2,3]. There is significant interest in understanding the dynamics of fluctuations in emotional state over time and how these changes differ among different people [4]. While self-reported longitudinal data over a relatively short time period [5] provides insights into emotion state and fluctuations, they only serve as a proxy for actual feelings. An alternative approach involves examining emotions through language usage. For instance, when experiencing happiness, people tend to use more happiness-associated words than usual, whereas during moments of anger, they are likely to use more anger-associated words [6]. Although humans experience many emotions, several psychological studies highlight the importance of a select few [7,8], e.g., the set of six Ekman emotions (*anger*, *disgust*, *fear*, *joy*, *sadness*, and *surprise*)

[7] or the set of eight Plutchik emotions that include Ekman's six emotions, as well as *trust* and *anticipation*. In this work, we focus on Plutchik's [8] eight emotions. For each emotion, we explore the extent to which that emotion exists across time: more trust or less trust. That is, when we explore the degree of trust, we explore the lack of trust (distrust/mistrust), the maximum amount of trust, and everything in between. This framework motivates our longitudinal analysis of shifting patterns in social media vaccine discourse.

**Social cognition theory:** We also draw ideas from social cognition theory and social psychology to understand the vaccine discourse on social media. Social cognition theory [9] argues that people judge other people or social groups based on two key dimensions: *warmth* (which relates to friendliness, trustworthiness, sociability, fondness, reverence) and *competence* (which relates to ability, power, dominance, and assertiveness) [10–14]. This framework is said to have developed because of evolutionary pressures — early humans needed to quickly assess whether someone was a friend (warm/positive) or foe (cold/negative) and whether they were competent (powerful) or incompetent (weak). This theory is widely used to study inter-group perceptions, for example, how people from one country perceive individuals from other countries or how certain groups perceive homeless, immigrants, and so forth [15]. Essentially, these two dimensions – *warmth* and *competence* – can be used to create a 4-quadrant space to analyze how different target groups are perceived. Recently, the social cognition theory has been used widely to study not just perceptions of people and groups but also other subjects, such as brands [16]. In this work, we apply the theory to study the perception of English-speaking X users towards vaccines. Thus, we examine the social media discourse on vaccines along the two dimensions of social cognition theory, which can be summarized as: good–bad (*warmth*) and competent–incompetent (*dominance*). Such analysis provides insights into how language (stance, warmth/competence cues) towards the vaccines are framed in X and whether people believe they are effective or ineffective in achieving those outcomes (where one is located in this $2 \times 2$ competence–warmth space may determine what kind of public messaging is suitable for them). It should be acknowledged that perception towards individual vaccines can vary, with prominent vaccines sometimes shaping public opinions. In this work, we focus on the perception of English-speaking X users towards vaccines in general, although it would be interesting to explore sentiment towards individual vaccines. From our initial examination of the data, over 75% of the English-language vaccine posts on X during COVID-19 years were related to COVID-19 vaccines.

We segment our analysis into two time periods: pre-COVID-19 vaccine discourse (2013–2019) and COVID-19 vaccine discourse (2020-2022) to examine how the pandemic has shifted vaccine-related discussions. To explore these shifts systematically, we focus on the following research questions (RQs):

*RQ1.* To what extent do English-language posts on X that mention vaccines use words associated with various emotions? How have the emotion word patterns changed before and during the COVID-19 pandemic years?

*RQ2.* From a social cognition perspective, how has the English-language discourse on vaccines on X changed over the years? Specifically, has there been an increase in the use of positive and warm words (suggesting growing trust and acceptance), or do we observe more competence-related words (highlighting perceptions of effectiveness and usefulness)?

*RQ3.* What approximate proportion of posts indicate a favorable stance towards vaccines, and what approximate proportion expresses opposition or skepticism? How have these proportions changed over time, especially in response to the COVID-19 pandemic?

*RQ4.* To what extent is vaccine opposition or skepticism characterized by untrustworthiness (low warmth) versus language that questions vaccine effectiveness (low competence)?

We focus on textual discourse rather than social media engagement metrics (likes, comments, and re-posts). While such engagement metrics may offer additional context, they do not directly capture how vaccines are framed within the social cognition dimensions. Our research aims to study vaccine discussions, rather than who discusses them or how

content spreads. Engagement metrics can be ambiguous, as a *like* could signal agreement, sarcasm, or even disagreement in some cases. By focusing on linguistic content, we ensure our findings are directly linked to the framing and perception of vaccines, which is central to public discourse analysis. We study the emotions and stances at the population-level across thousands of English-language X posts, rather than classifying individual posts. For such aggregate analysis, lexicon-based methods are found to be empirically near-optimal while offering greater simplicity, interpretability, and substantially lower computational and carbon footprint compared to advanced models. Teodorescu & Mohammad [17] show that when aggregating a few hundred instances per bin, lexicon-based methods produce very high correlation with the gold arcs (0.98 at bin = 100), and the incremental gains from machine learning (e.g., Transformer models) at such aggregation levels are minimal.

The four research questions systematically examine English-language vaccine discourse on X. RQ1 identifies and tracks emotion-related language. RQ2 builds on this by using social cognition theory to assess shifts towards more positive, competence-focused discourse. RQ3 focuses on users' stance, measuring support or criticism, while RQ4 explores negative discourse, distinguishing between emotional opposition and skepticism about effectiveness. This structured approach moves from general emotional trends to theoretical framing, explicit stance, and the nuances of skepticism, providing a comprehensive view of evolving vaccine perception.

To study these questions, we evaluate the emotional content of vaccine-related posts on X by examining the average frequency of emotional words used in posts from the pre-COVID-19 period (2013–2019) versus posts from the COVID-19 period (2020–2022). Using a Large Language Model (LLM) to assess stances towards vaccines, we analyze how discussions vary between pro-vaccine and anti-vaccine stances. We find that both positive and negative emotions increased during the pandemic, with notable rises in *surprise*, *trust*, and *anticipation*, reflecting heightened emotional responses and evolving perceptions in English-language X posts about vaccines. At the beginning of COVID-19, the discourse shifted towards more positive and competence-focused language but became more polarized with increased anti-vaccine sentiments towards the end of the pandemic. Anti-vaccine posts consistently used more negative words, highlighting persistent skepticism despite overall increased emotional engagement.

## Related works

The analysis of social media posts, and especially posts from X, has been widely used for opinion mining for understanding public perception and opinions toward vaccines. Hu et al. [18] looked into public opinion and perception of vaccines in the United States using spatiotemporal posts during the COVID-19-specific period. D'Andrea et al. [19] used an automated system to infer trends in public opinion regarding the stance towards the vaccination topic in Italian posts. A stance dataset towards vaccines during COVID-19 from posts in French, German, and Italian languages was collected by Giovanni et al. [20]. Similarly, various works carried out the sentiment, opinion analysis on COVID-19 vaccines from social media posts, including countries like India and Italy [21–23]. Works on emotion analysis from Stella et al. [24] looked into emotion profiling of posts from Italian users during the COVID-19 pandemic lockdown for studying public feelings during unexpected events, whereas Alhuzali et al. [25] studied the emotions and topics expressed on X during COVID-19 in the United Kingdom. More recent works curated data and studied sentiment, hesitancy, and dynamics towards vaccines during the COVID-19 [26–33]. Poddar et al. [34] studied the discourse of posts specific to anti-vaccines from 2018 to 2023. Osuji et al. [35] conducted a U.S.-based survey to analyze the public perception of COVID-19 vaccine safety. S1 Table provides additional details of works on sentiment and emotion analysis along with topic modeling and stance detection on COVID-19 vaccine-related social media posts.

Lexical analysis has been applied in a wide variety of fields and adapted to languages other than English. For example, Aggarwal et al. [36] used a valence-arousal-dominance framework to study the emotional expression differences between males and females from social media Reddit posts involving emotionally charged discourse during COVID-19. Hipson et al. [37] used a computational linguistic approach to analyze posts crawled using terms such as "solitude," "lonely," and

"alone" to understand how different people experience such emotions and the choice of language to discuss their experiences of these emotions. Numerous studies have shown that the dimensions of warmth and competence have profound implications across a wide range of domains, including, interpersonal status [38], social class [39], self-beliefs [40], political and cultural perception [41,42], child development [43], and organizational processes such as hiring, employee evaluation, and resource allocation [44]. Research indicates that warmth significantly influences the valence of interpersonal judgment, determining whether impressions are perceived as positive or negative.

Warmth is a core dimension in social perception and is often conceptualized as a subset of valence [45,46]. While valence captures a general emotional tone–positive or negative–warmth relates to perceived intent and encompasses two key facets: sociability (friendliness, likability) and trustworthiness (honesty, integrity) [47,48]. Together, these facets shape the judgment of a target's intentions, making warmth central to social and emotional evaluations.

While many studies focus exclusively on the COVID-19 period or specific emotions (and thus lack a comprehensive emotional analysis), our research examines over a decade of vaccine-related posts on X, leveraging human-annotated lexicons to assess sentiment and opinion shifts before and during the COVID-19 pandemic. Specifically, we analyze how emotional language associated with vaccines has evolved, investigating changes in the discourse from a social cognition perspective. We also study the proportion of posts indicating a favorable or skeptical stance towards vaccines and how these proportions have shifted over time, particularly during the COVID-19 pandemic. Finally, we assess the language that vaccine skeptics use, with a focus on low-competence language.

## Data collection

Using X's full archive search API before they stopped academic API access on February 2023, we collected historical public posts for 10 years from the start of 2013 (2013/01/01) to the end of 2022 (2022/12/31), consisting of 7 years of posts before COVID-19 (2013–2019) and 3 years during the COVID-19 pandemic (2020–2022). We only collected English-language posts (excluding re-posts) using vaccine-related keywords. The specific keywords used are: *vaccine, vaccines, vaccinates, vaccinate, vaccinated, vaxxed, vaccination, vaccinations, #vaccine, #vaccines, #vaccinate, #vaccinated, #vaccination, #vaccinations, #vaccinated, #vaxxed, #vaccinate*. Fig 1A shows the volume of actual posts available to our search query (using X's counts API that allows retrieving the total posts count for a given query without actually collecting the posts) and the volume of posts that we crawled, while Fig 1B shows the Monthly Active Users (MAU) and monetizable Daily Active Users (mDAU) gathered from official U.S. Securities and Exchange Commission (SEC) annual filings from Twitter before it went private. Twitter moved from reporting MAU to mDAU in April 2019. We find a significant rise in the total number of vaccine-related posts in the years following the start of the COVID-19 pandemic. Similarly, user engagement metrics show a gradual increase in MAU between 2013 and early 2019, followed by a steeper growth trajectory in mDAU from 2019 through the end of 2022.

After collecting the historical vaccine-related posts, we applied several preprocessing steps, as follows. We kept one post per user per day since the volume of posts we crawled was very large and also to reduce the influence of prolific user accounts on our analysis. We removed re-posts, emoticons, URLs, non-ASCII characters, and user mentions (e.g., @username) from the posts. Manual inspection of sample posts revealed that some of the crawled posts were related to a music band called "The Vaccines." To identify such posts, we used a Sentence-Transformer [49] fine-tuned on a large corpus of text to create embeddings for both the posts and the phrase "The Vaccines music band." We then filtered out posts based on their cosine similarity with the embedding of the phrase "The Vaccines music band," retaining only those with a similarity score smaller than 0.7. Approximately 2% of the total posts were filtered out. To assess the accuracy of this filtering procedure, we manually annotated a random sample of 200 posts (100 from the excluded set and 100 from the retained set). Treating references to the band as the positive class, 97% of the filtered out posts were correctly identified as referring to the music band (false-positive rate = 3%), while none of the retained posts referenced the band (false-negative rate = 0%), indicating a high level of classification precision.

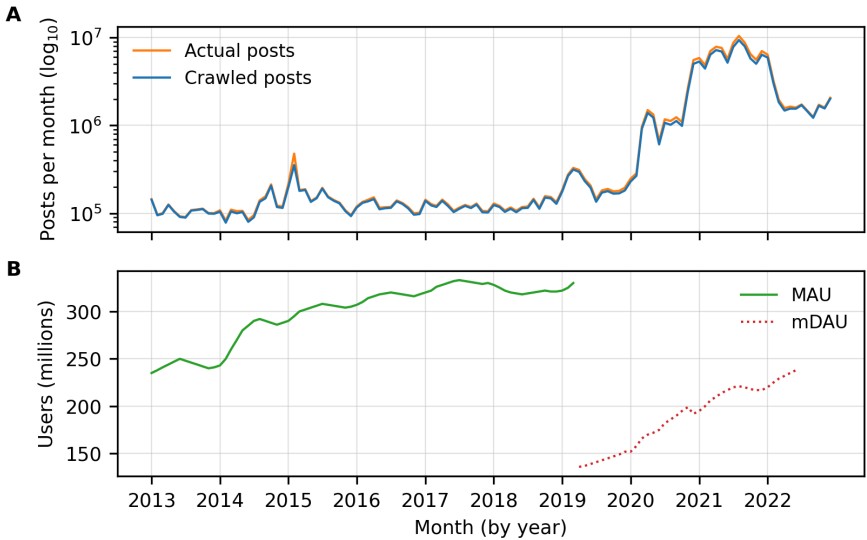

**Fig 1**. **Temporal trends in vaccine-related posts and platform engagement activities from 2013-2022.** (A) Monthly volume of vaccine-related posts is shown in $log_{10}$ scale. The orange line represents the total number of vaccine-related tweets posted ("actual posts"), and the blue line represents the tweets crawled by full-archive search ("crawled posts"). (B) Platform-level engagement metrics are plotted for the same period. The green line shows Twitter's Monthly Active Users (MAU), and the dotted red line represents monetizable Daily Active Users (mDAU), based on SEC filings prior to privatization. Twitter shifted from reporting MAU to mDAU in April 2019.

We collected a total of 129M posts from 13M unique users. After preprocessing, we removed 85.59% of the total posts and 56.65% of the total unique users, leaving us with 18,730,502 posts and 5,872,259 unique users. This rigorous filtering ensures that our dataset is not only large in scale but also of high quality, making it a valuable resource for both current analysis and future research endeavors. Fig 2 shows the distribution of total words per post and total characters per post. The majority of posts contain 14-16 words, whereas the distribution of character counts per post is spread out, with the majority of posts being around 100-110 characters long. The dataset will be made available for research purposes to facilitate progress on tools that can help understand the vaccine discourse on social media.

As a control, we also curated posts containing medical terms from 2015 to 2021 using the TUSC-Dataset [50]. TUSC-Dataset contains more than 38 million posts from the US and Canada from 2015 to 2021. From this dataset, we selected tweets that contain any of the keywords: *medicine*, *meds*, *medicines*, *antibiotic*, *antibiotics*, resulting in 100,000 tweets from 2015 to 2021.

## Methodology

We employ the Utterance Emotion Dynamics (UED) metric framework [51] that traces the emotional arc associated with utterances along the temporal axis. We rely on the NRC Emotion Lexicon [52,53], and the Words of Warmth Lexicon [54, 55] to determine the UED metrics. To detect stance towards vaccines, we employ large language models (LLMs), specifically a Llama model. The UED metrics, emotion lexicons as well as the process used for LLM-based stance detection are described in what follows.

### Utterance Emotion Dynamics (UED) Metrics

**Home Base**: The home base is defined as a subspace of high-probability emotional state space where a speaker is most likely to be found [56] and can be utilized to analyze utterances. For instance, at any given point in an individual's narrative, their location in the warmth–competence space may be determined by computing the averages of the warmth and

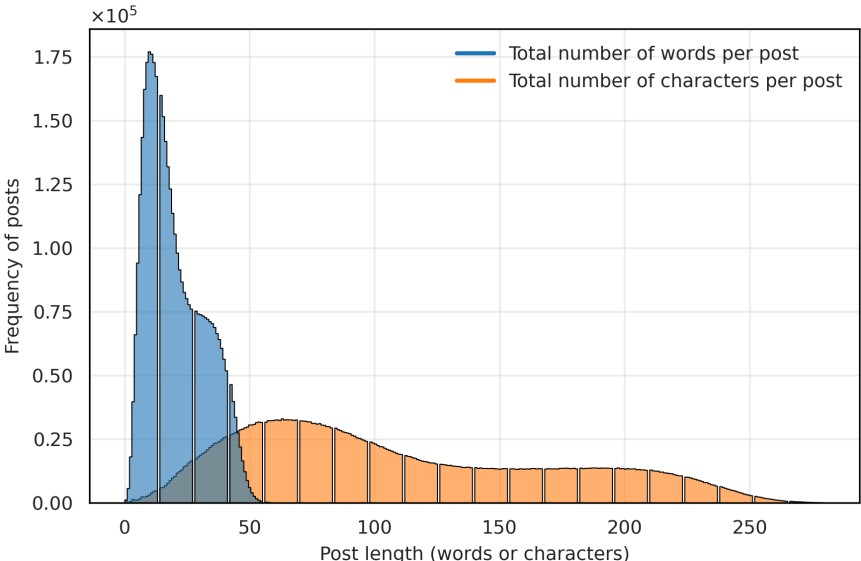

**Fig 2**. **Distribution of word count and character count per post.** The x-axis shows the count of words or characters, and the y-axis shows the frequency of posts in units of 100,000.

competence scores of the words within a small window of utterance. The trajectory followed by this position over time represents the individual's emotional arc or trajectory. The home base is defined as the subspace where the individual is most likely to be located.

In the one-dimensional case, for example, warmth ($w$) or competence ($c$), the home base refers to the subspace associated with the most common average warmth or competence scores, respectively [56]. Mathematically, this band is defined as the lower and upper bounds of a confidence interval, as shown below for warmth:

$$\overline{w} \pm t_{(1-\alpha, N-1)} \sqrt{\frac{\sigma^2}{N}} \tag{1}$$

where $\overline{w}$ is the mean of $w$, $t$ is the t-distribution, $N$ is the number of components in $w$, $\alpha$ is the desired confidence (e.g., 68% –one standard deviation away from the mean), and $\sigma^2$ is the variance of the warmth values in $w$.

In the two-dimensional warmth–competence space, the home base is bounded within an ellipse defined by:

$$\frac{w_i - \overline{w}}{\psi \lambda_1} + \frac{c_i - \overline{c}}{\psi \lambda_2} = 1 \tag{2}$$

where $\overline{w}$ and $\overline{c}$ represent the means for warmth and competence, respectively, $\psi$ denotes the critical $\chi^2$ value for the desired confidence range, while $\lambda_1$ and $\lambda_2$ represent the eigenvalues of the covariance matrix. The values in the denominator of the two terms correspond to the major and minor axes, representing the two diameters of the ellipse. The coordinates $w$ and $c$ that satisfy the equality in Eq (2) are the boundaries of the ellipse, whereas any set of coordinates for which the expression on the left of Eq (2) is smaller than 1 are located within the ellipse. Consequently, we can compare the home base ellipses among different individuals (or groups) in terms of their location in the warmth–competence space and their size based on the major and minor axes of the ellipses.

**Emotional Variability (EV)**: Emotional variability measures how a speaker's emotional state changes over time. According to Krone et al. [57], in a one-dimensional case, EV is defined as the standard deviation (SD) of the dimension considered, as shown below for warmth:

$$SD(w) = \frac{\sum_{i=1}^{N}(w_i - \overline{w})^2}{N} \tag{3}$$

In the two-dimensional case, EV is defined as the average of the standard deviations of $w$ and $c$. The above metrics collectively capture the temporal emotional characteristics of an individual's utterances. A more detailed discussion of these metrics and their use for lexical analysis has been provided by Hipson et al. [51]. We performed a similar lexical analysis utilizing the UED framework with warmth and dominance (competence) dimensions to study emotional expressions and opinions about vaccines, both in general over the 10 years and in the periods before and during the COVID-19 pandemic, using posts from the years 2013 to 2022.

## Emotion Lexicons

The UED metrics described above are determined using two existing word-emotion association lexicons: (1) the NRC Emotion Lexicon [52,53] and (2) the Words of Warmth Lexicon [55].

The NRC Emotion Lexicon comprises about 14,000 commonly used English words and their associations with eight emotions (*anger*, *anticipation*, *disgust*, *fear*, *joy*, *sadness*, *surprise* and *trust*), along with two sentiments (*positive* and *negative*) that are annotated manually using crowd-sourcing. The Words of Warmth Lexicon includes a list of more than 26,000 English words, each assigned a normalized score between 0 and 1 along the arousal, dominance (competence), trust, sociability, and warmth dimensions.

As we used vaccine-related words to crawl the posts in our dataset to study the shifting landscape of English-language vaccine discourse on X, we removed the entries in the lexicon that are morphological variants of the word *vaccine* (specifically, *vaccine* and *vaccination*). We also removed entries related to physical and mental illness (e.g., flu, polio, amnesia, etc.). This is to avoid the potential bias caused by the high frequency of these variants in our dataset.

It is important to note that the lexicons provide probable emotion associations without considering the contextual influence of the neighboring words within the target text. However, given that the majority of words usually possess a dominant primary sense [58], and the metrics capture emotion associations from a large number of words, this approach proves to be effective.

## Stance detection

LLMs have been effective for stance detection in social media posts due to the vast amount of data they are trained on [59]. To identify the stance of posts about vaccines, we use the Meta's Llama 3.3 instruction-tuned model with 70 billion parameters [60] and provide a prompt that asks the model to classify each post with respect to inferred stance of the poster towards vaccines as "favor," "against," or "neither of the two inferences can be reasonably made" (see S1 File for the exact prompt). Due to the large volume of posts in our dataset, we randomly sampled 2,000 posts per month across all years (a total of 240,000 posts) to be annotated by the LLM, as annotating the entire dataset would be very expensive both in terms of time as well as computational resources.

Our study goes beyond basic quantitative analysis by applying a theoretically grounded social cognition theory to examine how vaccines are perceived along the dimensions of warmth (positive-negative emotions) and competence (perceived effectiveness). In addition, we utilize LLMs for longitudinal tracking of vaccine-related shifts from 2013–2022, identifying key trends before and during the COVID-19 pandemic. Using LLMs, we conduct scalable stance detection to classify attitudes toward vaccines and analyze emotional tone and perceived efficacy to understand changes in vaccine trust. By integrating social cognition theory, LLM-based stance detection, and longitudinal analysis, our study offers a deeper

understanding of the evolving English-language vaccine discourse on X and could potentially have important implications for public health communication in English.

## Results and discussion

In this section, we discuss the results and findings of our research questions.

*RQ1. To what extent do English-language posts on X that mention vaccines use words associated with various emotions? How have the emotion word patterns changed before and during the COVID-19 pandemic years?*

Fig 3 shows the emotion-word density of positive and negative sentiment words from the NRC Emotion Lexicon per month across various years. Darker lines show the three-month rolling average of emotion-word density, and lighter lines represent the observed monthly values. Emotion-word density is calculated as the total number of emotion-related words used in a month divided by the total number of words posted in that month. We find that, during the pre-COVID-19 years, the negative word usage varied more significantly as compared to the positive word usage, and English-speaking X users generally used more negative words as compared to positive words. Unlike any of the pre-COVID-19 years, the usage of negative words dropped significantly during early 2021, even below the usage of positive words, when vaccines were being developed and released for the public (see S2 File for the COVID-19 vaccine development timeline). However, we see a significant divergence from late 2021 onwards, where the usage of negative words increased rapidly, while positive word usage saw a declining trend. This may suggest vaccine hesitancy and discussions about the side effects of COVID-19 vaccines on X social media. The trend towards more negative language highlights challenges faced by public health officials and scientists in maintaining public trust in vaccines.

Fig 4 shows the monthly trends in emotion-word density during the COVID-19 years (2020-2022). The emotion-word density is calculated as the total number of emotion-related words used in a month divided by the total number of words posted in that month. Fear starts high in early 2020 and drops significantly at the beginning of 2021 before gradually increasing again in late 2021 and throughout 2022. This mirrors the trajectory of the pandemic: initial fear of the unknown,

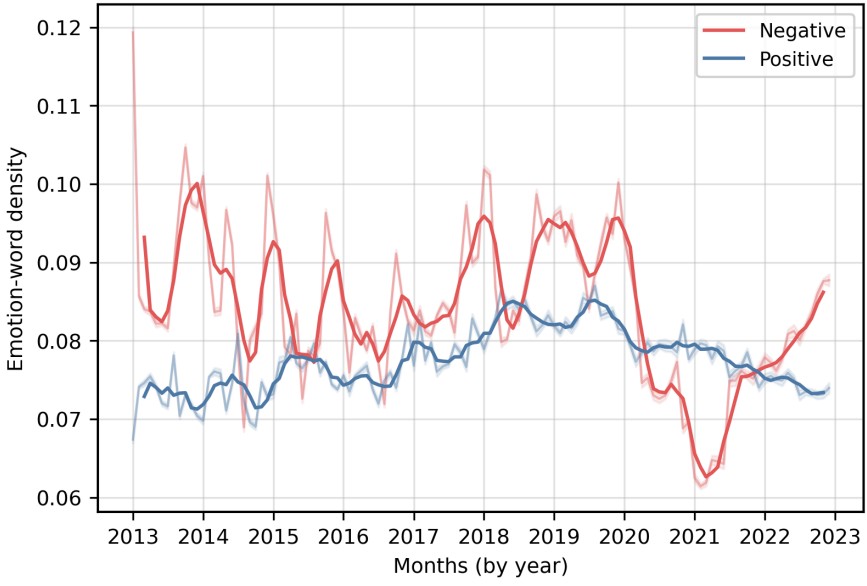

**Fig 3**. **Monthly trend in emotion-word density of positive and negative sentiment words from 2013 to 2022.** Darker lines show the three-month rolling averages of emotion-word density, and lighter lines represent the observed monthly values. Emotion-word density is calculated as the total number of emotion-related words used in a month divided by the total number of words posted in that month.

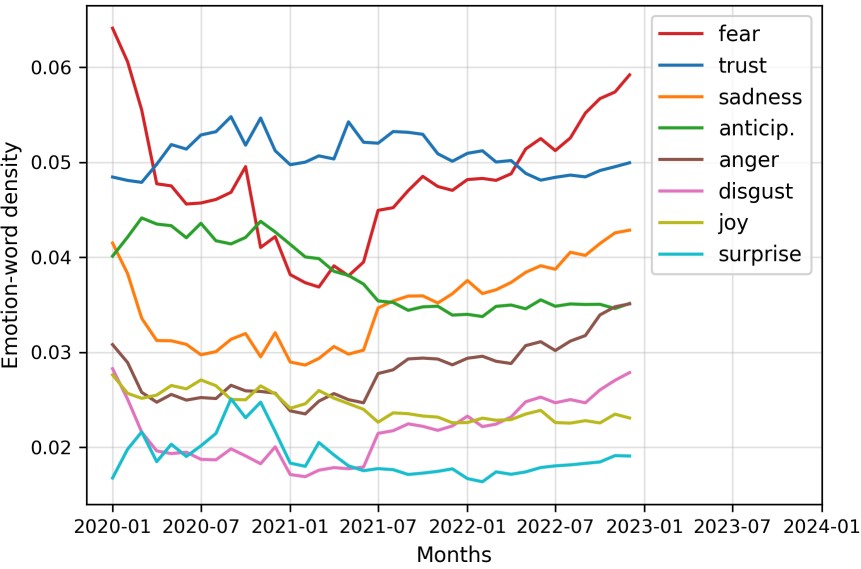

**Fig 4**. **Monthly trends in emotion-word density during the COVID-19 pandemic (2020–2022).** Emotion-word density is calculated as the total number of emotion-related words used in a month divided by the total number of words posted in that month.

followed by more information and control measures, including the approval of COVID-19 vaccines, then renewed concerns with variants and long-term impacts of vaccines. Anticipation was higher at the beginning of 2020 but gradually decreased over 2021 and 2022, suggesting people were highly anticipating the vaccines, but the anticipation decreased as vaccines became available after 2021. Other emotions like anger, disgust, joy, surprise, and trust remained relatively stable, suggesting these were not the dominant emotional responses to the pandemic.

Table 1 compares the emotion-word density of various emotions across all pre-COVID-19 years (2013–2019) and COVID-19 years (2020–2022). As before, emotion word density is calculated as the total number of emotion-related words used in a month divided by the total number of words posted in that month. The mean and standard deviation for each emotion are calculated using monthly word densities in pre-COVID-19 years and COVID-19 years. Negative emotion word usage decreased during the COVID-19 period, by 13.22% (from 0.0876 to 0.0760). However, we didn't find a statistically significant change in positive emotion word usage. We saw significant increase ($p < 0.001$) in surprise (from 0.0156 to 0.0190, i.e. 21.66%), and trust (from 0.0459 to 0.0508, i.e. 10.64%). However, other emotions showed a decrease in COVID-19 years, notably, disgust (which decreased from 0.0301 to 0.0217, i.e., 27.81%) and fear (which decreased from 0.0632 to 0.0481, i.e., 23.89%), and also sadness (which decreased from 0.0406 to 0.0348, i.e., 14.23%), joy (which decreased from 0.0276 to 0.0243, i.e., 11.87%), and anger (which decreased from 0.0292 to 0.0281, i.e., 3.85%). Although the mean value of anticipation emotion increased, the result was not statistically significant.

To conclude the analysis related to RQ1, the overall decrease in negative emotions suggests that vaccine discussions became slightly less negative during the pandemic. Higher usage of trust words suggests growing confidence in vaccines and expectations surrounding their development and distribution. The rise in surprise emotions could be related to rapid developments in vaccine research, policy changes, or unexpected events during the pandemic. The decrease in joy words could signal a dampened expression of happiness during the crisis. Finally, minimal change in positive and anticipation emotions could imply that expression of positivity and forward-looking sentiment remained relatively stable despite the upheaval. Overall, these trends suggest that while general negativity decreased, emotional expression was diversified– marked by greater trust and surprise–capturing both adaptation and uncertainty towards vaccines in the COVID-19 years.

**Table 1. Emotion-word density of various emotions during pre-COVID-19 (2013–2019) and COVID-19 (2020–2022) periods, with percent change (% Change), and p-values (Mann-Whitney U test).** Emotion-word density is calculated as the total number of emotion-related words used in a month divided by the total number of words posted in that month.

| Emotion | Pre-COVID-19 | | COVID-19 | | % Change | p-value |
|---|---|---|---|---|---|---|
| | Mean | SD | Mean | SD | | |
| Negative | 0.0876 | 0.0085 | 0.0760 | 0.0079 | -13.22 | <0.001 |
| Positive | 0.0777 | 0.0047 | 0.0770 | 0.0025 | -0.84 | 0.5344 |
| Anger | 0.0292 | 0.0023 | 0.0281 | 0.0031 | -3.85 | <0.05 |
| Anticipation | 0.0371 | 0.0020 | 0.0381 | 0.0037 | 2.65 | 0.5766 |
| Disgust | 0.0301 | 0.0045 | 0.0217 | 0.0032 | -27.81 | <0.001 |
| Fear | 0.0632 | 0.0076 | 0.0481 | 0.0067 | -23.89 | <0.001 |
| Joy | 0.0276 | 0.0019 | 0.0243 | 0.0015 | -11.87 | <0.001 |
| Sadness | 0.0406 | 0.0047 | 0.0348 | 0.0044 | -14.23 | <0.001 |
| Surprise | 0.0156 | 0.0018 | 0.0190 | 0.0021 | +21.66 | <0.001 |
| Trust | 0.0459 | 0.0039 | 0.0508 | 0.0019 | +10.64 | <0.001 |

*Note:* Percent change is based on the difference in mean values between COVID-19 and pre-COVID-19 periods, calculated as ((COVID-19 mean – Pre-COVID-19 mean)/Pre-COVID-19 mean) × 100. Values with $p < 0.05$ and $p < 0.001$ are considered statistically significant.

*RQ2. From a social cognition perspective, how has the English-language vaccine discourse on X changed over the years?*

From a social cognition perspective, the discourse on vaccines has shifted over time, specifically during the COVID-19 pandemic. Fig 5 compares "home bases" for pre-COVID-19 and COVID-19 periods. We observe that these ellipses have different shapes: the pre-COVID-19 ellipse (red) is less widespread in the competence space, as compared to the COVID-19 ellipse (blue). This suggests that during pre-COVID-19, there was not much polarization around competence: that is,

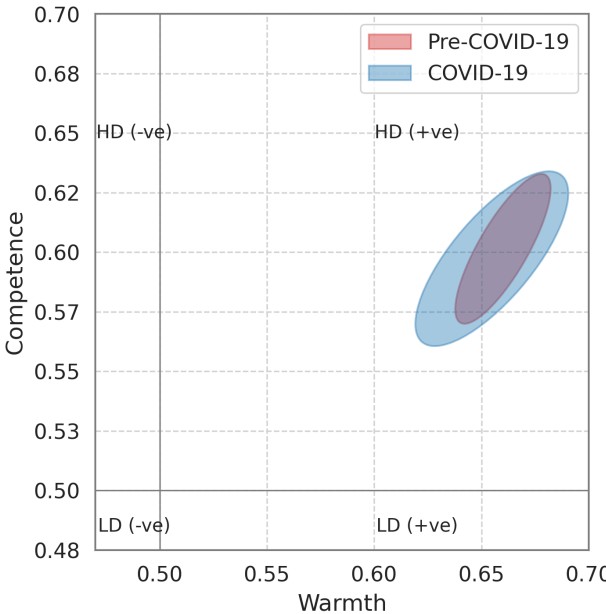

**Fig 5. Comparison of vaccine-discourse "home bases" before and during COVID-19.** The red ellipse represents home bases for pre-COVID-19 years and the blue during the COVID-19 years in the warmth–competence space.

discussions around the effectiveness/competence of the vaccines were relatively stable. However, there was marked variability around competence during the COVID-19 years.

Fig 6 shows the comparison of warmth and competence dimensions. Fig 6A shows changes in warmth and its two distinct components - trust and sociability - over time from 2013 to 2022. We find a stable trend in warmth during pre-COVID-19 periods, whereas the usage of warm words increased in the COVID-19 period from 2020 to early 2021 before decreasing consistently until the end of 2022. We see a similar trend in the trust and sociability aspect of the warmth dimension. Interestingly, the usage of the warmth words remained consistent in control tweets containing medical terms, even during the COVID-19 period. This suggests that the public trust in vaccines may have increased during the early stages of the pandemic, but declined once vaccines became readily available. Several factors could have contributed to this shift, including concerns about the rapid development and roll-out of the vaccines. Fig 6B shows the change in competence over time. We find that competence increased during the first year of COVID-19 before decreasing consistently after early 2020. In the control medical tweets, we find that competence remains consistent during the COVID-19 years. This decrease in competence suggests heightened uncertainty and a lack of control. As the pandemic progressed, people shifted towards using more passive or cautious language. Additionally, public discourse often emphasized vulnerability, empathy, and shared hardship, which could have further contributed to the use of less competence-related words.

*RQ3. What approximate proportion of posts indicate a favorable stance towards vaccines, and what approximate proportion expresses opposition or skepticism? How have these proportions changed over time, especially in response to the COVID-19 pandemic?*

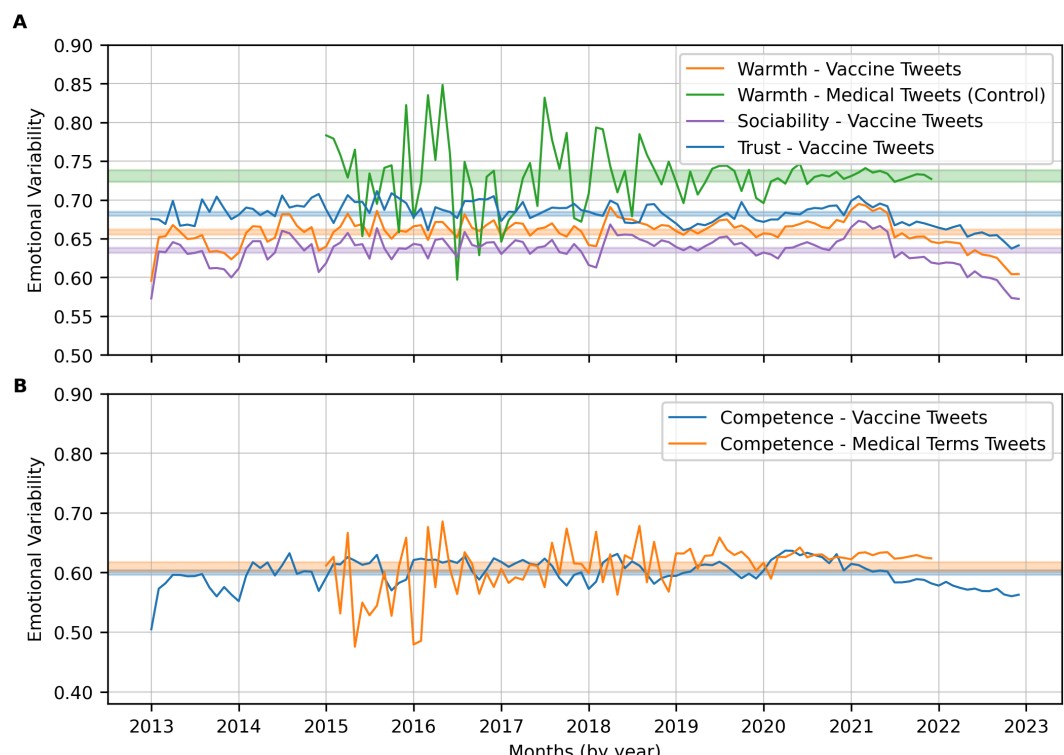

**Fig 6**. **Trends in warmth (trust, sociability) and competence language in vaccine discourse from 2013 – 2022.** **(A)** Warmth and its two components—trust and sociability—over time. **(B)** Competence over time. Medical-term tweets serve as a control. The shaded bands represent the home bases for each dimension.

As mentioned above, we used the Llama 3.3 model to answer this question. To assess the model's accuracy, we manually annotated 500 posts and used the model to predict those posts 5 times. The average agreement between Llama 3.3 annotations and human annotations over the 5 runs was 66.32%, with a standard deviation of $\pm$ 1.94%. Note that random guessing will result in an accuracy of 33.33%. Table 2 shows the per-class precision, recall, and F1-score for each stance category, along with overall accuracy and averaged metrics for classification using the Llama 3.3 model. The model showed strong performance in identifying both "favor" and "against" posts with high recall, indicating reliable detection of clear positive and negative stances. The temperature parameter in large language models (LLMs) modulates output randomness by shaping the sampling distribution. Lower temperatures produce more deterministic text, whereas higher values enhance variability and creativity by increasing the likelihood of less probable tokens [61]. We tested temperature values of 0.0, 0.4, 0.7, and 1.0 using 500 manually annotated posts, yielding accuracies of 66.32% $\pm$ 2.01%, 66.32% $\pm$ 1.94%, 66.36% $\pm$ 2.14%, and 66.40% $\pm$ 2.19%, respectively. As we did not find a significant difference in accuracy by varying the temperature parameter, we adopted 0.4 for the experiments.

While the majority of our dataset is labeled using the Llama model, this manual annotation of 500 samples serves as a quality check, providing a measure of confidence in the model's performance.

Table 3 shows some example posts together with the annotations by the Llama 3.3 model, compared against corresponding human annotations. Some of the instances where the Llama 3.3 model made mistakes were posts with aggressive and negative words; or posts where the stance is not explicitly stated and rather subtle, as seen in the last two posts in Table 3.

Since the classifier's accuracy (66.32%) at post-level shows moderate agreement with human judgment on a sample of 500 posts and is above the accuracy of the random baseline (33.33%), it can be used to approximate directional changes, specifically, whether the proportion of posts against vaccines has increased or decreased at a monthly aggregated level. This allows us to approximately identify broad trends in vaccine attitudes across the population over the years. However, these estimates should not be treated as precise proportions or as a representative of the general population.

**Table 2**. **Per-class and global average performance metrics for stance classification obtained from 5 independent runs of Llama 3.3 model on 500 (manually annotated) posts, using a temperature of 0.4.** Precision, recall, and F1-scores are reported for each class label. Accuracy, "macro avg" and "weighted avg" are reported globally for the set of 500 posts. "Macro avg" indicates the unweighted mean of the per-class metrics, while "weighted avg" accounts for class imbalance. Accuracy is used as a proxy for the agreement between Llama 3.3 annotations and human annotations on 500 sample posts.

| Class | Precision | Recall | F1-score | Support |
|---|---|---|---|---|
| against | 0.5464 $\pm$ 0.0299 | 0.9282 $\pm$ 0.0144 | 0.6875 $\pm$ 0.0241 | 141 |
| favor | 0.7458 $\pm$ 0.0187 | 0.8775 $\pm$ 0.0284 | 0.8062 $\pm$ 0.0211 | 192 |
| neutral | 0.8087 $\pm$ 0.0307 | 0.2254 $\pm$ 0.0211 | 0.3520 $\pm$ 0.0258 | 167 |
| accuracy | – | – | 0.6632 $\pm$ 0.0194 | 500 |
| macro avg | 0.7003 $\pm$ 0.0220 | 0.6770 $\pm$ 0.0058 | 0.6152 $\pm$ 0.0133 | 500 |
| weighted avg | 0.7158 $\pm$ 0.0179 | 0.6632 $\pm$ 0.0217 | 0.6170 $\pm$ 0.0214 | 500 |

**Table 3**. **Examples of posts labeled by Llama 3.3 model (llama_label) and human annotators (human_label).**

| Posts | llama_label | human_label |
|---|---|---|
| The autism one really gets me … considering it's a genetic defect that you're born with and not something transmitted through vaccines. | in-favor | in-favor |
| Parents: heard the new vaccine requirement? Meningitis vaccine required for IL students entering 6th, 12th grade. | neither | neither |
| Then it's not recommended. the govt is making the public suffer by gambling with this, i cant. Make the non-medical govt officials take the vaccine first. | against | against |
| "One of the last holy grails of HIV research is the development of a HIV vaccine." #homenews #news #cryptonews | neither | in-favor |
| Watching the age go up for what's considered eligible for the vaccine. KNOCK IT OFF | against | in-favor |

Fig 7 shows the approximate proportion of posts in favor and the proportion against vaccines over the years. We observe that there is a slight increasing trend of estimated share of posts in favor of vaccines between 2013 and 2020. Starting from 2020 onwards, we see three trends of substantial decrease, substantial increase back to pre-2020 levels, and finally, a substantial decrease in such posts. The drop in 2020 is likely because this phase was marked by posts that talk about the lack of, the need for, and the difficulty of developing a new vaccine for COVID-19. By late 2020, with the news of successful testing of new COVID-19 vaccines, we observe a growth of in-favor posts. However, with the deployment of vaccines in 2021, a general decrease in the virus virulence, and greater awareness of vaccines, the percentage of in-favor posts has steadily declined.

The percentage of against-vaccine posts also has a slight increasing trend from 2013 onwards, which lasts until about 2020. There is, however, a notable decline in such posts in late 2020, and from 2021 onwards, we observe a sharp increase in the percentage of against posts. Notably, by late 2022, our LLM-based estimates suggest that the percentage of posts against vaccines has surpassed those in favor of vaccines. The sharp increase in anti-vaccine stance after 2021 correlates with the introduction of COVID-19 vaccines, their rapid development, and later concerns about their safety and efficacy. The COVID-19 pandemic created a surge of anxiety, uncertainty, and polarized opinions around vaccination, potentially leading to a marked decline in pro-vaccine sentiment and a rise in skepticism and opposition.

*RQ4. To what extent is vaccine opposition or skepticism characterized by untrustworthiness (low warmth) versus language that questions vaccine effectiveness (low competence)?*

Table 4 shows the emotion-word density for low warmth and low competence words used per post during pre-COVID-19 (2013–2019) and during COVID-19 (2020–2022) years by English-speaking X users with in-favor and against stances towards vaccines. Emotion-word density is calculated as the total number of emotion-related words divided by the total number of words per post. We observed significant temporal shifts in the emotional tone of vaccine-related discourse from the pre-COVID-19 period to the COVID-19 era. Among posts that were in favor of vaccines, emotion word density associated with low warmth (i.e., untrustworthiness) decreased from 0.1006 to 0.0942, representing a 6.4% decrease (Mann-Whitney U test, $p < 0.001$). Similarly, the density of words associated with low competence emotions declined from 0.0698 to 0.0688, corresponding to a 1.4% decrease (Mann-Whitney U test, $p < 0.001$). In contrast, posts expressing an anti-vaccine stance exhibited an increase in low warmth emotion word density, rising from 0.1199 to 0.1214 (a 1.3% increase; Mann-Whitney U test, $p < 0.001$), while low competence emotion-word density declined from 0.0742 to 0.0725 (a 2.5% decrease; Mann-Whitney U test, $p < 0.001$). These findings suggest a polarization in emotional framing, with pro-vaccine

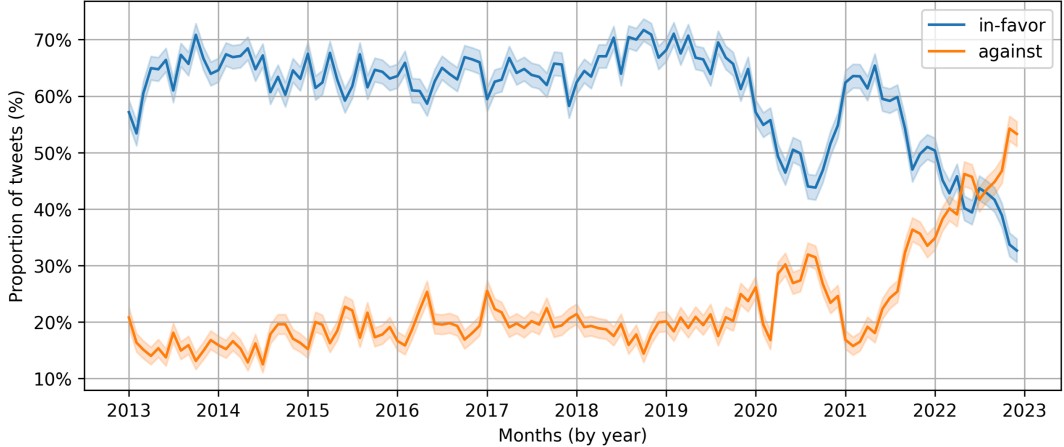

**Fig 7. Monthly proportion of pro- and anti-vaccine posts from 2013–2022.** For each month, 2,000 posts were sampled and classified as in-favor or against vaccination.

**Table 4. Emotion-word density for low-warmth and low-competence emotions.** Emotion-word density is calculated as the proportion of emotion-related words to the total number of words in each post. Percent change expresses the percent increase or decrease from the pre-COVID period to the COVID period, calculated as ((COVID-19 mean - pre-COVID-19 mean)/pre-COVID-19 mean) x 100%.

| Emotion | Pre-COVID-19 (2013–2019) | | COVID-19 (2020–2022) | | % change | |
|---|---|---|---|---|---|---|
| | In favor | Against | In favor | Against | In favor | Against |
| Low warmth | 0.1006 | 0.1199 | 0.0942 | 0.1214 | -6.4% | +1.3% |
| Low competence | 0.0698 | 0.0743 | 0.0688 | 0.0725 | -1.4% | -2.5% |

discourse becoming less negatively charged, whereas anti-vaccine discourse demonstrated a modest intensification of low warmth emotional expression.

To gain more insights into this question, we also performed a tree-map analysis of the top 15 words with low warmth and low competence. The tree-maps for the top 15 words with low warmth for in-favor and against vaccine stances are shown in S1 Fig while the tree maps for the top 15 words with low competence are shown in S2 Fig. We find that pro-vaccine stance posts include top words like "die", "bad", "risk", and "shot", focusing on concerns/risks and negative outcomes, along with words like "wear" (referring to mask) and "shot" focusing on preventative actions, whereas anti-vaccine stance posts include top words like "bad", "dangerous", "forced" and "mandatory", suggesting concerns about choice of vaccination. The anti-vaccine stance posts use more emotionally charged negative words, potentially aiming to evoke fear. The pro-vaccine stance, while also using negative terms, focuses more on risks and preventative actions. Similarly, pro-vaccine stance posts use low competence words like "ill", "die", "masks", and "wait", whereas anti-vaccine stance posts use words like "fake", "fear", "injury", and "dead." The anti-vaccine stance posts show more words related to distrust (e.g. "fake", "fear") and lack of personal agency (e.g. "forced", "mandatory"), highlighting their key concerns.

## Conclusions

In this work, we applied the social cognition theory to study the English-language vaccine discourse on X (formerly Twitter) over the past decade, specifically before and during the COVID-19 pandemic. Our analysis demonstrates that vaccine discourse became more emotionally charged during the pandemic, with decreases in negative emotion expressions. The early discourse during the COVID-19 pandemic reflected optimism and trust in vaccine development. However, this was later followed by polarization towards the end of COVID-19 years. These observations pertain to English-language posts on X and are viewed as corpus-level patterns.

Furthermore, our study highlights the potential of utilizing a large-scale, high-quality dataset to examine public vaccine discourse. Spanning over a decade, this dataset offers rich opportunities for future research by leveraging cutting-edge LLM-based techniques, such as self-supervised, semi-supervised, and active learning–to better understand public sentiment and improve the detection of stance and emotion in public health communication.

## Limitations and future research

The scope of this study is limited to English posts on X from 2013 to 2022 that mention vaccines. Thus, the conclusions apply to this dataset and should not, on their own, be used to draw conclusions about people at large. It is known, for example, that X users tend to be younger and technologically savvy compared to the overall population. Most of the X data is not geo-tagged, so we cannot determine the extent to which these posts come from various world regions. Furthermore, the analysis focuses on English-language posts, which may not reflect global trends or cultural differences in vaccine perceptions. Lastly, while the study covers a significant timeframe, it may not capture long-term trends beyond the scope of the data collection period which was limited to a decade between January 1st, 2013 and December 31st, 2022. That said, since X is such an influential platform, it is useful to better understand the discourse on vaccines. Future research could address these limitations by incorporating data from multiple social media platforms, improving stance detection results, analyzing non-English content, and extending the timeframe of the study.

As shown in this research, LLMs are able to accurately determine the stance of people towards vaccines in a majority of cases. However, it should be noted that people use language in subtle and nuanced ways (including sarcasm and humor); social media posts do not always provide all the necessary context to understand individual posts; and there is considerable person-to-person difference in how we use language. Thus, while the use of LLMs to determine broad trends in vaccine stance is reasonable, they should not be used on their own to draw conclusions about individual posters.

Our analysis is focused exclusively on textual discourse and does not incorporate social media engagement metrics such as likes, comments, and re-posts. Although these signals could provide insights into content diffusion, influence, and reach, they fall outside the scope of this study. By focusing exclusively on linguistic content, we ensure that our findings are directly tied to the framing and perception of vaccines within public English-language discourse on X. However, this omission constrains our ability to draw conclusions about how widely particular narratives spread or how they shape broader audience engagement patterns.

Using SEC data for monthly trend analysis requires interpolation to estimate monthly trends because Twitter only reported user statistics quarterly (every three months). Therefore, such interpolation may not accurately reflect actual month-to-month fluctuations, seasonal patterns, or event-driven user behavior changes. Additionally, the methodological breakdown between MAU (2013-2019) and mDAU (2019-2022) reporting creates a fundamental discontinuity that prevents direct comparison of user growth trends across the platform's most significant transition period.

While our primary focus is on understanding the shifts in vaccine discourse before and during the COVID-19 pandemic, an important question still remains: Will vaccine discourse return to its pre-pandemic state, or has the landscape permanently shifted? Due to limitations in data collection under the new policy of X, our dataset only extends through the end of 2022. Examining post-pandemic trends would provide valuable insights into the long-term impact of COVID-19 on public perceptions of vaccines. Future research could explore whether the emotionally charged discourse during the pandemic persists, or if the discourse has stabilized over time.

## Supporting information

**S1 Fig. Top 15 Low-Warmth Words by Stance.** Treemap visualizations of the fifteen low-warmth words in the vaccine-related posts for **(A)** in-favor stance and **(B)** against stance.
(TIFF)

**S2 Fig. Top 15 Low-Competence Words by Stance.** Treemap visualizations of the fifteen low-competence words in the vaccine-related posts for **(A)** in-favor stance and **(B)** against stance.
(TIFF)

**S1 Table. Summary of related works on vaccines and COVID-19 post datasets.** The related works are grouped based on their main topics–"Sentiment Analysis", "Emotion Analysis", "Topic Modeling", "Stance Detection", and "Vaccine misinformation and opinion mining".
(PDF)

**S1 File. Prompt used for Llama 3.3 model.** The prompt with "system" and "user" message was sent to the Llama 3.3 model to get stance classification.
(DOCX)

**S2 File. COVID-19 vaccines timeline from 2020 to 2022.** The file contains the vaccine timeline from COVID-19 virus discovery to vaccine development and boosters. Source: Wikipedia.
(DOCX)

## Author contributions

**Conceptualization:** Nikesh Gyawali, Doina Caragea, Cornelia Caragea, Saif M. Mohammad.

**Data curation:** Nikesh Gyawali.

**Funding acquisition:** Doina Caragea.

**Investigation:** Nikesh Gyawali, Doina Caragea.

**Methodology:** Nikesh Gyawali, Doina Caragea, Cornelia Caragea, Saif M. Mohammad.

**Project administration:** Doina Caragea.

**Resources:** Nikesh Gyawali.

**Software:** Nikesh Gyawali, Saif M. Mohammad.

**Supervision:** Doina Caragea, Cornelia Caragea, Saif M. Mohammad.

**Visualization:** Nikesh Gyawali, Saif M. Mohammad.

**Writing – original draft:** Nikesh Gyawali.

**Writing – review & editing:** Doina Caragea, Cornelia Caragea, Saif M. Mohammad.

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
