## [Decision Letter · Decision Letter 0]

4 Sep 2025

PONE-D-25-27112The Shifting Landscape of Vaccine Discourse: A Massive Dataset Examining a Decade of Social Media Posts on Vaccines Before and During the COVID-19 PandemicPLOS ONE

Dear Dr. Caragea,

Thank you for submitting your manuscript to PLOS ONE. After careful consideration, we feel that it has merit but does not fully meet PLOS ONE’s publication criteria as it currently stands. Therefore, we invite you to submit a revised version of the manuscript that addresses the points raised during the review process. In particular, I have summarized the issues raised by the reviewers as follows.

The most important concerns relate to the methodological robustness of your emotion and stance detection. Reviewer 1 noted that the use of the NRC lexicon requires stronger justification, and suggested either providing a clear rationale for this choice or considering the use of more advanced transformer-based methods. Reviewer 2 raised concerns about the stance detection results, observing that the agreement between the Llama 3.1 model and human annotators (around 66%) is too low to support precise quantitative claims. These outputs should therefore be framed as exploratory or approximate rather than definitive. In the same area, Reviewer 1 also asked for fuller reporting of model performance (precision, recall, and F1 scores per class) and clarification of the temperature settings used in your prompting strategy.

Another point raised by Reviewer 2 concerns the scope and framing of your findings. At times, the paper suggests broader implications for public perceptions or societal views, but since your dataset is limited to English-language posts on X/Twitter, conclusions should be carefully framed as such rather than generalized to the population at large. Relatedly, Reviewer 2 encouraged a clearer articulation of the theoretical framework, or at least an acknowledgment of its speculative nature if it cannot be fully justified.

The reviewers also highlighted data quality and limitations. Reviewer 2 asked for some manual validation of your filtering procedure to ensure that irrelevant posts (such as references to “The Vaccines” band) were not mistakenly included or excluded. They also pointed out that your decision to omit engagement metrics such as likes, shares, and retweets is reasonable, but should be more explicitly presented as a limitation since it constrains conclusions about diffusion, influence, and reach.

Finally, Reviewer 1 recommended improvements to figures and measures. Specifically, greater clarity is needed in labeling (for example, using “number of characters” rather than “length” and adding missing y-axis labels). Figure 1 would be more useful if contextualized against overall Twitter activity, while Figure 3 and Table 3 would benefit from using the proportion of emotion words rather than average word count, or at least from visually marking the 2017 change in Twitter’s character limit to avoid misinterpretation.

In sum, the main revisions needed are to (1) strengthen and clarify methodological choices (emotion detection, stance detection, dataset filtering), (2) temper the generalization of findings, (3) address theoretical framing, and (4) improve figure clarity and limitations discussion.

We look forward to receiving your revised manuscript.

Kind regards,

Carlos Carrasco-Farré

Academic Editor

PLOS ONE

Journal Requirements:

Reviewers' comments:

Reviewer's Responses to Questions

**Comments to the Author**

1. Is the manuscript technically sound, and do the data support the conclusions?

Reviewer #1: Partly

Reviewer #2: Yes

2. Has the statistical analysis been performed appropriately and rigorously?

Reviewer #1: Yes

Reviewer #2: Yes

3. Have the authors made all data underlying the findings in their manuscript fully available?

Reviewer #1: Yes

Reviewer #2: Yes

4. Is the manuscript presented in an intelligible fashion and written in standard English?

Reviewer #1: Yes

Reviewer #2: Yes

5. Review Comments to the Author

Reviewer #1: I would like to thank the authors for their submission. I appreciate how comprehensive this paper is in its view of vaccine discourse on Twitter/X over the years, and I find the results about shifts in emotional language and the emotional content of anti-vax posts to be particularly interesting. Overall, I think the paper is nearly ready for publication -- I just have a few notes about how the paper could be revised/improved.

My primary concern with the paper is the use of the NRC lexicon as a method of emotion detection. It's possible that this is the best method to use for your case, but I don't see a strong justification for it written in the paper. My sense is that transformer-based supervised learning methods would be better for the task of emotion recognition. Therefore, I would either like to see a more accurate method used for emotion detection, or a justification for why the lexicon should be used over deep learning methods.

Another, less important, concern I have is that Figure 1 would be more useful if there were some baseline representing how much overall activity on Twitter has increased over the years (e.g., is the spike in 2015 due to a bunch more people using Twitter or were people really more likely to talk about vaccines at that time?) I understand that these data might be hard to come by now since the API is no longer accessible, but if you're able to find some dataset representing overall Twitter activity during your period of study, it would be useful to include it as a comparison point.

Similarly, for Figure 3 and Table 3 -- is average number of words really the best measure here? I would think that measuring the average proportion of emotion words would be better, especially because of the fact that Twitter increased their character limit. The huge spike attributable to that increase in late 2017 makes it difficult to compare any time before that to any time after that. If you do choose to stick to number of words instead of relative proportion, I would encourage you to include a vertical line in Figure 3 at the point in time when Twitter increased their character limit (the figure could be circulated without the caption).

In Figure 2, I would also change the word "length" to "number of characters," since the meaning of the word "length" is ambiguous (could also apply to number of words).

I've also noticed that y-axis labels are missing from many of these plots, make sure to add those.

Finally, I see that you report the accuracy of your stance detection model -- what are the precision, recall, and f1 scores for each class? I also think the temperature of the language model matters a lot for your evaluation setup, where you prompt the model 5 times for each post. If the temperature is too low, it will give you the same output every time. If it's too high, it might be too random. What temperature value did you set for the model, and how would your average/std. error of accuracy vary for different temperature values?

Reviewer #2: The manuscript presents a large-scale analysis of vaccine-related discourse on X/Twitter between 2013 and 2022. The dataset is impressive in scope and will be useful for future research. The integration of lexical analyses and LLM-based stance classification is timely. However, in its current form, the manuscript overstates its contributions and requires important revisions before it can be considered acceptable.

Major concerns

1/ Reliability of data. The reported agreement between Llama 3.1 and human annotators (66.2% ±4.6) is too low to support precise quantitative claims. It can be used for identifying broad trends, it cannot provide accurate proportions of pro/anti vaccine discourse. The manuscript must explicitly frame these outputs as exploratory and approximates. The claims suggesting exact distributions of results should be removed or softened.

2/ Overgeneralization of results. The paper often refers to "public perceptions" or "societal views." This is misleading: the dataset only covers English-language posts from a single platform, X, which is demographically skewed. Conclusions should consistently be limited to "English-language vaccine discourse on X" rather than generalized to the population at large.

3/ Theoretical framework. The manuscript should provide stronger theoretical justification or acknowledge the speculative nature of their application.

4/ Dataset filtering. The procedure for excluding irrelevant content (i.e., references to "The Vaccines" band) relies on embeddings and thresholds, but no validation is reported. Without a sample-based check, the risk of false positives/negatives remains high. The authors should report at least some manual validation to strengthen confidence in dataset integrity.

5/ Engagement metrics. The exclusion of likes, shares, and retweets is justified as ambiguous, but this decision limits interpretability. The limitations section should more explicitly state that the absence of engagement analysis prevents conclusions about diffusion, influence, or reach of vaccine discourse.

---

## [Author Response · Author response to Decision Letter 1]

17 Oct 2025

Manuscript title: The Shifting Landscape of Vaccine Discourse: A Massive Dataset Examining a Decade of English-language Social Media Posts on Vaccines Before and During the COVID-19 Pandemic

Dear PLOS One Editor,

We would like to thank you and the reviewers for your valuable comments and constructive suggestions, which have helped us improve our manuscript. Below, we provide a detailed, point-by-point response to each comment. Revisions are indicated in the manuscript using highlighted text in blue for insertion and strike-through for removal.

Comments from Academic Editor:

Comment 1:

The most important concerns relate to the methodological robustness of your emotion and stance detection. Reviewer 1 noted that the use of the NRC lexicon requires stronger justification and suggested either providing a clear rationale for this choice or considering the use of more advanced transformer-based methods.

Response:

We appreciate bringing up this concern. We have revised the manuscript to clarify our methodological rationale as follows:

“We study the emotions and stances at the population-level across thousands of posts, rather than classifying individual posts. For such aggregate analysis, lexicon-based methods are found to be empirically near-optimal while offering greater simplicity, interpretability, and substantially lower computational and carbon footprint compared to advanced models. Teodorescu & Mohammad (2023) show that when aggregating a few hundred instances per bin, lexicon-based methods produce very high correlation with the gold arcs (0.98 at bin=100), and the incremental gains from the ML/Transformer model at such aggregation levels are minimal.”

This clarification has been incorporated into the revised manuscript (lines 116–124 in the marked-up version).

Comment 2:

Reviewer 2 raised concerns about the stance detection results, observing that the agreement between the Llama 3.1 model and human annotators (around 66%) is too low to support precise quantitative claims. These outputs should therefore be framed as exploratory or approximate rather than definitive.

Response:

We have revised the manuscript to moderate the claims and explicitly characterize the stance detection results as exploratory and approximate, consistent with the reviewer’s recommendation. Furthermore, we re-ran the analysis using the updated Llama 3.3 model (replacing Llama 3.1); however, the overall performance and observed trends remained effectively unchanged.

Comment 3:

In the same area, Reviewer 1 also asked for fuller reporting of model performance (precision, recall, and F1 scores per class) and clarification of the temperature settings used in your prompting strategy.

Response:

In response to the reviewer’s comment, we have added a new table (Table 3 in the marked-up manuscript) reporting the model’s per-class performance metrics, including precision, recall, and F1 scores. We have also clarified the temperature settings used in the LLM prompting strategy (lines 453–464 in the marked-up version).

Comment 4:

Another point raised by Reviewer 2 concerns the scope and framing of your findings. At times, the paper suggests broader implications for public perceptions or societal views, but since your dataset is limited to English-language posts on X/Twitter, conclusions should be carefully framed as such rather than generalized to the population at large.

Response:

We have revised the manuscript to more clearly delineate the scope and applicability of our findings, ensuring that the conclusions are appropriately contextualized within the dataset’s limitations. Specifically, we now emphasize that our analyses are based solely on English-language posts from X (formerly Twitter) and that the results reflect trends within this corpus rather than generalizable conclusions about global or population-wide attitudes. These clarifications have been incorporated in the abstract, introduction, and conclusion sections (lines 16, 140, and 540 in the marked-up version), as well as expanded in the Limitations and Future Research section (lines 554–568), where we discuss representational constraints and avenues for future cross-lingual and cross-platform validation.

Comment 5:

Relatedly, Reviewer 2 encouraged a clearer articulation of the theoretical framework, or at least an acknowledgment of its speculative nature if it cannot be fully justified.

Response:

We have expanded the theoretical framing in the Introduction (lines 35–89 in the marked-up manuscript) to provide a clearer articulation of the two complementary theoretical perspectives—emotion dynamics and social cognition theory—that guide our analysis. The revised text explicitly explains how each framework informs the study: emotion dynamics theory motivates the longitudinal examination of emotional fluctuations over time, while social cognition theory offers a structured lens for interpreting warmth and competence dimensions in vaccine discourse. We also clarify that, while these frameworks provide interpretive grounding, their application to large-scale social media data is exploratory in nature, acknowledging the theoretical and methodological boundaries of such an approach.

Comment 6:

The reviewers also highlighted data quality and limitations. Reviewer 2 asked for some manual validation of your filtering procedure to ensure that irrelevant posts (such as references to “The Vaccines” band) were not mistakenly included or excluded.

Response:

We have strengthened the manuscript by explicitly reporting the manual validation of our data filtering procedure, as suggested. Specifically, to verify that irrelevant posts referenced to “The Vaccines music band” were accurately excluded, we manually annotated a random sample of 200 posts (100 excluded and 100 retained). Treating references to the band as the positive class, 97% of the excluded posts were correctly identified as referring to the band (false-positive rate = 3%), and none of the retained posts referenced the band (false-negative rate = 0%), confirming high precision in our filtering. These details have been incorporated into the revised manuscript (lines 223–228 in the marked-up version).

Comment 7:

They also pointed out that your decision to omit engagement metrics such as likes, shares, and retweets is reasonable, but should be more explicitly presented as a limitation since it constrains conclusions about diffusion, influence, and reach.

Response:

We have revised the Limitations and Future Research section (lines 576–583 in the marked-up manuscript) to explicitly acknowledge that our analysis focuses exclusively on textual content and omits engagement metrics such as likes, comments, and reposts. The updated text clarifies that, while this decision allows us to concentrate on linguistic framing and perception of vaccines, it limits our ability to draw conclusions about content diffusion, influence, and audience reach. We now explicitly frame this omission as a methodological limitation, noting that future work could integrate engagement metrics to better capture the spread and impact of vaccine-related discourse.

Comment 8:

Finally, Reviewer 1 recommended improvements to figures and measures. Specifically, greater clarity is needed in labeling (for example, using “number of characters” rather than “length” and adding missing y-axis labels).

Response:

We have revised all figures to improve clarity and consistency in labeling. Specifically, we replaced ambiguous terms (e.g., “length”) with more precise descriptors such as “number of characters” and added missing y-axis labels to ensure readability and interpretability across all visualizations.

Comment 9:

Figure 1 would be more useful if contextualized against overall Twitter activity

Response:

In response to the reviewer’s comment, we have enhanced Figure 1 by contextualizing vaccine-related posting activity against overall Twitter engagement trends. Specifically, we incorporated Monthly Active Users (MAU) and monetizable Daily Active Users (mDAU) metrics obtained from official SEC filings, providing a broader view of platform-level activity over the study period. Furthermore, we have explicitly discussed the limitations of these metrics in the Limitations and Future Research section (lines 584–590), noting that (1) Twitter reported MAU data until 2019 and switched to mDAU thereafter, which introduces a discontinuity, and (2) interpolation of quarterly SEC data to estimate monthly trends may obscure short-term fluctuations and event-driven behaviors.

Comment 10: while Figure 3 and Table 3 would benefit from using the proportion of emotion words rather than average word count, or at least from visually marking the 2017 change in Twitter’s character limit to avoid misinterpretation.

Response: Thank you for the helpful suggestion. In the revision, we re-computed and reported emotion trends using emotion-word density (i.e., the proportion of emotion words) rather than average word counts. Accordingly, Figure 3 and the corresponding table (now Table 1 in the revised manuscript) present monthly emotion-word density, defined as the total number of emotion-associated words used in a month divided by the total number of words posted that month.

Reviewers' comments:

Reviewer #1 Comment:

My primary concern with the paper is the use of the NRC lexicon as a method of emotion detection. It's possible that this is the best method to use for your case, but I don't see a strong justification for it written in the paper. My sense is that transformer-based supervised learning methods would be better for the task of emotion recognition. Therefore, I would either like to see a more accurate method used for emotion detection, or a justification for why the lexicon should be used over deep learning methods.

Response:

We study the emotions and stances at the population-level across thousands of posts, rather than classifying individual posts. For such aggregate analysis, lexicon-based methods are found to be empirically near-optimal while offering greater simplicity, interpretability, and substantially lower computational and carbon footprint compared to advanced models. Teodorescu & Mohammad show that when aggregating a few hundred instances per bin, lexicon-based methods produce very high correlation with the gold arcs (0.98 at bin=100), and the incremental gains from the ML/Transformer model at such aggregation levels are minimal.

Reviewer #1 Comment:

Another, less important, concern I have is that Figure 1 would be more useful if there were some baseline representing how much overall activity on Twitter has increased over the years (e.g., is the spike in 2015 due to a bunch more people using Twitter or were people really more likely to talk about vaccines at that time?) I understand that these data might be hard to come by now since the API is no longer accessible, but if you're able to find some dataset representing overall Twitter activity during your period of study, it would be useful to include it as a comparison point.

Response:

Thank you for this helpful suggestion. In the revision, we now provide a platform‐level baseline in Figure 1B by plotting Twitter’s Monthly Active Users (MAU) and monetizable Daily Active Users (mDAU) extracted from official SEC filings. We note in the text that Twitter transitioned from MAU to mDAU reporting in April 2019, and we use the two series accordingly. Alongside Figure 1A (vaccine-related post volume), these metrics allow readers to contextualize fluctuations in vaccine discourse against overall platform growth. We also describe the observed pattern--gradual MAU growth from 2013 to early 2019 and a steeper mDAU trajectory from 2019-2022--in the Data Collection section.

Because SEC user counts are reported quarterly, we interpolated to monthly values for visualization. We explicitly caution that this interpolation may not capture true month-to-month dynamics and that the MAU to mDAU change introduces a discontinuity that limits direct comparison across the boundary; these caveats are stated in the manuscript’s Limitations. These changes are reflected in lines 205-208 and 599-605 in the marked-up manuscript.

Reviewer #1 Comment:

Similarly, for Figure 3 and Table 3 -- is average number of words really the best measure here? I would think that measuring the average proportion of emotion words would be better, especially because of the fact that Twitter increased their character limit. The huge spike attributable to that increase in late 2017 makes it difficult to compare any time before that to any time after that. If you do choose to stick to number of words instead of relative proportion, I would encourage you to include a vertical line in Figure 3 at the point in time when Twitter increased their character limit (the figure could be circulated without the caption).

Response:

Thank you for this thoughtful suggestion--we agree with the concern about post-length changes. In the revision, Figure 3 now reports emotion-word density (proportion of emotion words) rather than average word counts per tweet. We define emotion-word density as “the total number of emotion-related words used in a month divided by the total number of words posted in that month,” and we plot both the observed monthly values and a three-month rolling average.

Accordingly, Table 3 (now Table 4 in the revised marked-up manuscript) present emotion-word density, defined as the total number of emotion-words used divided by the total number of words used.

The corresponding changes in the text related to the Figure 3 and Table 4 are in lines 340-344 and 523-535.

Finally, for completeness, Table 1 in the revision also reports emotion-word density across emotions using the same definition. The corresponding changes in the text are marked in lines 392-397 and 409 to 415 in the marked-up revision manuscript

Reviewer #1 Comment:

In Figure 2, I would also change the word "length" to "number of characters," since the meaning of the word "length" is ambiguous (could also apply to number of words).

Changed the word “length” to “number of characters”

I've also noticed that y-axis labels are missing from many of these plots, make sure to add those.

Response:

Thank you for pointing this out. In the revision, we clarified the terminology in Figure 2 by referring to post “number of characters” (total characters) rather than the ambiguous “length,” and we updated the caption/text accordingly to state that post length is measured in total characters.

We also added y-axis labels across all figures. For example, Figure 2 now specifies that the y-axis shows the frequency of posts (in units of 100,000) and Figure 3 has y-axis label as word density.

Reviewer #1 Comment:

Finally, I see that you report the accuracy of your stance detection model -- what are the precision, recall, and f1 scores for each class? I also think the temperature of the language model matters a lot for your evaluation setup, where you prompt the model 5 times for each post. If the temperature is too low, it will give you the same output every time. If it's too high, it might be too random. What temperature value did you set for the model, and how would your average/std. error of accuracy vary for different temperature values?

Response:

Thank you for the helpful suggestion. In the revision, we:

1. Report per-class metrics: We added Table 2 with precision, recall, and F1-scores for each class, alongside overall accuracy and macro/weighted averages (see Table 2 and text at lines 452–455 in the marked-up manuscript).

2. Detail temperature settings and their effect: We tested temperatures 0.0, 0.4, 0.7, and 1.0 and report corresponding accuracies with standard deviations--66.32% +- 2.01%, 66.32% +-1.94%, 66.36% +- 2.14%, 66.40% +- 2.19%--finding no significant difference across temperatures; we therefore adopted 0.4—with lowest standard deviation--for experiments (lines 458–464).

Reviewer #2 Comment:

1/ Reliability of data. The reported agreement between Llama 3.1 and human annotators (66.2% ±4.6) is too low to support precise quantitative claims. It can be used for iden

---

## [Decision Letter · Decision Letter 1]

16 Nov 2025

The Shifting Landscape of Vaccine Discourse: A Massive Dataset Examining a Decade of  English-language Social Media Posts on Vaccines Before and During the COVID-19 Pandemic

PONE-D-25-27112R1

Dear Dr. Caragea,

We’re pleased to inform you that your manuscript has been judged scientifically suitable for publication and will be formally accepted for publication once it meets all outstanding technical requirements.

Kind regards,

Carlos Carrasco-Farré

Academic Editor

PLOS ONE

Additional Editor Comments (optional):

Reviewers' comments:

Reviewer's Responses to Questions

**Comments to the Author**

1. If the authors have adequately addressed your comments raised in a previous round of review and you feel that this manuscript is now acceptable for publication, you may indicate that here to bypass the “Comments to the Author” section, enter your conflict of interest statement in the “Confidential to Editor” section, and submit your "Accept" recommendation.

Reviewer #1: All comments have been addressed

Reviewer #2: All comments have been addressed

2. Is the manuscript technically sound, and do the data support the conclusions?

Reviewer #1: Yes

Reviewer #2: Yes

3. Has the statistical analysis been performed appropriately and rigorously?

Reviewer #1: Yes

Reviewer #2: Yes

4. Have the authors made all data underlying the findings in their manuscript fully available?

Reviewer #1: Yes

Reviewer #2: No

5. Is the manuscript presented in an intelligible fashion and written in standard English?

Reviewer #1: Yes

Reviewer #2: Yes

6. Review Comments to the Author

Reviewer #1: (No Response)

Reviewer #2: The revised manuscript presents a clear, methodologically sound, and transparent investigation of the relationship between neuroticism and social media addiction, incorporating the mediating role of perceived algorithmic personalization. The authors have satisfactorily addressed all prior reviewer and editorial comments, improving structure, clarity, and theoretical framing.

The text is concise, logically organized, and free of overstatement. References and variable naming have been clarified. No ethical, methodological, or data integrity concerns are detected.

Nevertheless:

Ensure that the data/code availability statement is complete and accessible.

Verify that item examples or full wording of the new scale appear in the manuscript or as supplementary material.

7. PLOS authors have the option to publish the peer review history of their article (what does this mean?). If published, this will include your full peer review and any attached files.

Reviewer #1: No

Reviewer #2: **Yes: **David Badajoz-Dávila

---

## [Editor Report · Acceptance letter]

PONE-D-25-27112R1

PLOS One

Dear Dr. Caragea,

I'm pleased to inform you that your manuscript has been deemed suitable for publication in PLOS One. Congratulations! Your manuscript is now being handed over to our production team.

Kind regards,

on behalf of

Dr. Carlos Carrasco-Farré

Academic Editor

PLOS One